# Circadian typology is related to emotion regulation, metacognitive beliefs and assertiveness in healthy adults

**Juan Manuel Antúnez***

Department of Personality, Assessment and Psychological Treatment, University of Málaga, Málaga, Andalucía, Spain

* antunez@uma.es

**Data Availability Statement:** Data file is available in OSF. URL: (https://osf.io/nh5c4)

**Funding:** This study was supported by the University of Málaga open access publishing fund.

## Abstract

Circadian typology has been related to several mental health aspects such as resilience, perceived well-being, emotional intelligence and psychological symptoms and disorders. However, the relationship between circadian typology and emotion regulation, metacognitions and assertiveness, which constitute core constructs related to psychological well-being and psychopathology, remain unexplored. This study aims to analyze whether circadian typology is related with those three constructs, considering the possible influence of sex. 2283 participants (833 women), aged 18–60 years (30.37 ± 9.26 years), completed the reduced Morningness-Eveningness Questionnaire, the Emotion Regulation Questionnaire, the Meta-Cognitions Questionnaire 30, and the Rathus Assertiveness Schedule. Main effects were observed between circadian typology and cognitive reappraisal, metacognitions, negative beliefs of uncontrollability and danger, cognitive confidence, cognitive self-consciousness, and assertiveness ($F_{(2,2276)} > 4.80$, $p < 0.009$, $\eta_p^2 > 0.004$, in all cases). Morning-type participants scored lower than evening-type in general metacognitive beliefs, negative beliefs of uncontrollability and danger, cognitive confidence, and cognitive self-consciousness, and higher than evening-type in cognitive reappraisal and assertiveness, while neither-type exhibited intermediate scores ($p < 0.033$ in all cases). According to the results, evening-type individuals might display a higher tendency to support maladaptive beliefs about thinking itself as well as a lesser tendency to reappraise a potentially emotion eliciting situations in order to modify its meaning and its emotional impact and to exert their rights respectfully. This new evidence improves the understanding of the relationships between circadian typology and psychological factors related to psychological well-being and psychopathology. Results implications for the onset and maintenance of psychological problems are discussed. Although future longitudinal studies are needed, results emphasize evening-type as a risk factor for the development of psychological disturbances and morning-type as a protective factor against those.

**Competing interests:** The author have declared that no competing interests exist.

## Introduction

During the last decades there has been an increase in the interest in studying the associations between morningness-eveningness and the so-called circadian typologies and mental health related-factors such as satisfaction with life and well-being [1,2], resilience and optimism [3], emotional intelligence [4], maladaptive coping strategies [5], personality traits related to risk behaviors [6], psychological symptoms [7,8] and psychopathology [9].

Morningness-eveningness is a dimension that tends to follow a normal distribution and allows for classifying the population in one-of-three circadian typologies or chronotypes: morning-, neither-, and evening-type [10]. The first ones are used to go to bed and wake up earlier, are more adjusted with the night-dark cycle and, therefore, show a phase advance of their biological and behavioral circadian functions as compared to evening-type, who tend to go bed and wake up later and are little adjusted with the light-dark cycle. Regarding prevalence, it is estimated that morning- and evening-type represents about 40% of adult population (around 20% for each one) while neither-type, which tends to maintain an intermediate position between extreme groups, compounds the remaining 60% [11].

The higher synchronization with the light-dark cycle in the morning-type may be the cause of the tendency to suggest this chronotype as a protective factor for the incidence of psychopathology [12], even suicidality [13], while evening-type has been suggested as a risk factor for the development of mental disturbances and psychological symptoms [9,11,14,15]. Those differences might be explained by the social jet-lag [16], which occurs when the social and the biological clocks are out of sync with each other.

Other relevant factors commonly related to circadian typology are age and sex. On the one hand, childhood and elderly populations have shown higher morning-type prevalence that contrasts with the evening-type predominance found in adolescents. After adolescence and during the adulthood it has been shown a slow but consistent shift toward morningness [17,18]. On the other hand, although some studies about circadian typology showed a greater tendency to morningness in women [19–21], even by assessing the circadian rhythmic expression [22], others have found the opposite [23–25]. In this line, psychosocial factors such as meal times, work schedule, family relationships and child care have been proposed as an explanation for this controversial [17].

Emotion regulation, known as the ability to regulate emotions in order to succeed, has been a widely studied subject during the last three decades [26] that has emerged as one of the pillars of well-being and social functioning [27–33]. According to a process model of emotion regulation, emotions may be regulated in five points along the emotion generative process: (1) selection of the situation, (2) modification of the situation, (3) deployment of attention, (4) change of cognitions (meanings) and (5) modulation of the experiential, behavioral or physiological responses. The first four points allow antecedent-focused emotion regulation, while the last allows response-focused emotion regulation [32]. Cognitive reappraisal and expressive suppression constitute two emotion regulation strategies commonly used in everyday life. The first, which is defined as the attempt to reappraise a potentially emotion-eliciting situation in a way that modifies its meaning and changes its emotional impact, is an antecedent-focused emotion regulation strategy and, due to its relationships with well-being, is considered an adequate strategy [27,32,34]. Expressive suppression, on the other hand, is defined as a form of response based on the inhibition of the emotion-expressive behavior and, therefore, is a response-focused emotion regulation strategy. Moreover, this strategy is more used by men and commonly categorized as maladaptive due to its relationship with depression and anxiety [27,32,33,35].

A field directly related with emotion regulation and psychopathology that has received an increased scientific attention during the last three decades has been metacognitions. In this line, although it is well known that emotions exert an important influence in cognitions [36], it is also known that metacognitions exert a central role for the development and persistence of emotional dysfunctions [37]. Metacognitions refer to the psychological structures, knowledge, events and processes involved in the control, modification and interpretation of thinking itself [38]. The self-regulatory executive function model [39] states that beliefs in psychopathology are the result of a metacognitive component that guides think and cope activity. In this sense, metacognitions lead to focus on disorder congruent information and to use maladaptive coping strategies [38], which is in line with the generic cognitive model [40]. Several studies using the Meta-Cognitions Questionnaire [38], which assess a range of metacognitive domains related with psychopathological processes such as low cognitive confidence, positive beliefs about worry, cognitive self-conscientiousness, negative beliefs about uncontrollability and danger, and belief concerning the need to control one's own thoughts, have shown a strong relationship between those and several psychological disorders and symptomatology. In this line, metacognitions have found to be related to different psychological issues such as depression [41], bipolar disorder [42], obsessive-compulsive disorder [43], generalized anxiety disorder [44], anxiety [45,46], gambling [47], substance use disorder [48] and eating disorders [49].

Assertiveness, known as the ability to affirm a right or a point of view without either aggressively threatening the rights of another nor submissively allows another to deny one's rights or point of view is another construct that has been found related to metacognitions [50]. This construct, which is considered as a continuum with excessive agreeableness (submissive or unassertive) and excessive hostility (aggressiveness) in the poles, has gained much attention from the psychotherapy field as its training is considered a valuable transdiagnostic intervention [51]. Several research studies have shown a link between assertiveness and clinical problems such as depression [52], anxiety [50,53,54], and serious mental illness [55], as well as with other constructs related to well-being such as self-esteem [54], relationship satisfaction [56,57], happiness [58] and meta-cognitive beliefs [50].

The aim of this work was to examine, for the first time, the relationships between circadian typology and three constructs related with psychopathology and well-being: Emotion regulation, metacognitions and assertiveness, considering the possible influence of sex, in a wide sample of healthy Spanish adults without physical or mental pathology. It is hypothesized that evening-type participants will score higher than morning-type in metacognitions and expressive suppression, while morning-type will score higher than evening-type in assertiveness and cognitive reappraisal.

## Materials and method

### Participants

The total amount of participants in the study was 3016, out of which 733 were excluded due to duplicate answers and for non-accomplishing the inclusion criteria (Spanish resident, absence of mental and physical health problems, absence of shift work and 18–60 yrs). The final sample included in the study was composed of 2283 adults, Spanish residents, aged 18–60 (30.37 ± 9.26 yrs), 1450 men (63.5%) and 833 women (36.5%). Age differences were observed between men (29.44 ± 8.39 yrs) and women (31.93 ± 10.34 yrs) ($t_{(2281)} = 6.25$; $p < 0.001$; Cohen's $d = 0.26$). Participants did not receive any payment for their participation in the study and they all gave their informed consent prior to the inclusion in the study. Subjects submitted their answers on online questionnaires of circadian typology, emotion regulation, metacognition beliefs and resilience and provided self-reported information about sociodemographic

variables and the presence of any mental or physical pathology. The present study, whose protocol was approved by the Research Committee of the University of Málaga, complied with the tenets of the Declaration of Helsinki and the international ethical standards of chronobiological research [59].

## Measurement instruments

The reduced Morningness-Eveningness Questionnaire (rMEQ), in its standardized version for the Spanish population [60], was used for the assessment of circadian typology. The rMEQ is composed of five items with total score ranging from 4 to 25 points. Participants are assigned to one of the three possible circadian typologies (morning-, neither-, or evening-type) according to the cutoff score: 4–11 for the evening-type, 12–17 for the neither-type, and 18–25 for the morning-type. The Spanish version of the rMEQ has been shown as a reliable measure for classifying individuals in the morningness-eveningness dimension [61] and its internal reliability was adequate for the present sample (Chronbach's α = 0.78).

Emotion regulation was assessed using the Spanish version of the Emotion Regulation Questionnaire (ERQ) [32,33], which is composed of ten items in a Likert scale format with scores ranging between 1 (strongly disagree) and 7 (strongly agree) for each one. The ERQ assesses two different emotion regulation strategies: Cognitive reappraisal, with 4 items and total score ranged 4–28, and expressive suppression, with 6 items and total score ranged 6–42. Higher score implies a higher use of the strategy. The ERQ shows good psychometric properties and is considered as a reliable and valid instrument for assessing emotion regulation strategies [32,33]. For the present sample, internal reliability was adequate for both strategies, with a Chronbach's α of 0.83 for expressive suppression and of 0.81 for cognitive reappraisal.

Metacognitions were measured using the Spanish version of the Meta-Cognitions Questionnaire 30 (MCQ-30) [38,62]. This questionnaire, which comprises five subscales, is composed of 30 items with a 4-point Likert scale response format, ranging from 1 (do not agree) to 4 (strongly agree). Total score ranges from 40 to 120 points and higher scores are indicative of greater pathological metacognitions. The five subscales assess positive beliefs about worry ("worry is useful for avoiding future problems"), negative beliefs of uncontrollability and danger ("my worrying might be dangerous for me"), cognitive confidence ("I do not trust my memory"), need to control thoughts ("I should be controlling my thoughts all of the time") and cognitive self-consciousness ("I am constantly aware of my thinking"). Internal reliability analyses with the present sample have shown adequate scores for the total score (Chronbach's α = 0.90) as well as for each subscale (Chronbach's α = 0.89 for positive beliefs about worry; 0.83 for negative beliefs of uncontrollability and danger; 0.86 for cognitive confidence; 0.73 for need to control thoughts; and 0.85 for cognitive self-consciousness).

Assertiveness was assessed using the Spanish version of the Rathus Assertiveness Schedule (RAS) [63,64], which is composed of 30 items in a 7-points Likert scale format ranging from -3 (very much unlike me) to 3 (very much like me), with total scores ranging from -90 to 90. RAS has been consolidated as a reliable and valid instrument for assessing assertiveness due to its adequate psychometric properties [65]. Likewise, internal reliability was adequate for the present sample (Chronbach's α = 0.86).

## Data analysis

Two multiple analyses of covariance (MANCOVA) and two single analyses of covariance (ANCOVA) were performed, each one with circadian typology and sex as factor and with ERQ, MCQ-30 subscales, RAS and MCQ-30 total scores as dependent variables, respectively, whereas age was taken as a covariate to control for possible effects. Post-hoc comparisons were adjusted

by Bonferroni's correction and the partial eta square $\eta_p{}^2$ was obtained as a measure of the ANCOVA effect size, considering 0.01 as small, 0.04 moderate, and 0.10 large. Three multiple stepwise regression analyses were performed, each one for scores on ERC, MCQ-30 and RAS as dependent variables, respectively, while sex and age (first step) and rMEQ scores (second step) were the independent variables. Cohen's $f^2$ was used as effect size, considering scores of 0.02 as small, 0.15 moderate, and 0.35 large [66]. Statistical analyses were performed using the IBM SPSS 64 bits software (version 24.0) and statistical tests were bilateral with the type I error set at 0.01.

# Results

## Sociodemographic data

Subjects distribution along circadian typology groups was 636 for the evening-type (27.9%; 452 men and 184 women), 1115 for the neither-type (48.8%; 716 men and 399 women), and 532 for the morning-type (23.3%; 282 men and 250 women). The distribution of the rMEQ scores were skewed towards the eveningness pole (z = 3.27, $p < 0.001$). Likewise, significant differences were observed between women (14.78 ± 0.15) and men (13.67 ± 0.11) in rMEQ scores ($t_{(1,2281)}$ = 5.89, $p < 0.001$, Cohen's $d$ = 0.25). Circadian typology groups showed age-differences ($F_{(2,2280)}$ = 105.31; $p < 0.001$). Post hoc comparisons showed differences between all groups. Morning-type participants were older (34.91 ± 0.46 yrs) than neither- (29.85 ± 0.26 yrs; $p < 0.001$) and evening-type (27.49 ± 0.30 yrs; $p < 0.001$), who were younger than neither-type ($p < 0.001$). Table 1 shows sociodemographic (employment and marital status) according to sex and circadian typology groups. A higher proportion of workers and a lower one of students was observed in the morning-type group while students and workers percentage in the evening-type were even. Regarding marital status, it was observed a higher prevalence of women as well as morning-type participants in the coupled group while evening-type were more prevalent in the single group.

## Emotion regulation

Descriptive data for the total sample as well as for sex and circadian typology groups in the Emotion Regulation Questionnaire is shown in Table 2. Significant main effects were observed for sex in cognitive reappraisal and expressive suppression while for circadian typology the significant main effect was in cognitive reappraisal. Post-hoc comparisons showed higher scores of cognitive reappraisal in morning-type as compared to evening-type participants (MD = 1.45, $p$ = 0.002, Cohen's $d$ = 0.20). Regarding sex, women showed higher cognitive reappraisal scores (MD = 0.83, $p$ = 0.008, Cohen's $d$ = 0.14) and lower expressive suppression scores (MD = 4.20, $p < 0.001$, Cohen's $d$ = 0.80).

## Metacognitions

Descriptive data for the total sample and structured by sex and circadian typology groups is shown in Table 2. Significant main effects were observed in MCQ-30 total score, negative beliefs of uncontrollability and danger, cognitive confidence and cognitive self-consciousness for circadian typology. According to post-hoc comparisons, neither-type participants showed higher scores than morning-type in cognitive confidence (MD = 0.70, $p$ = 0.012, Cohen's $d$ = 0.12) and scored lower than evening-type in MCQ-30 total score (MD = 1.94, $p$ = 0.032, Cohen's $d$ = 0.21) and cognitive self-consciousness (MD = 0.65, $p$ = 0.016, Cohen's $d$ = 0.23). Evening-type subjects showed higher scores than morning-type in MCQ-30 total score (MD = 3.13, $p$ = 0.002, Cohen's $d$ = 0.17), negative beliefs of uncontrollability and danger (MD = 0.91, $p$ = 0.002, Cohen's $d$ = 0.23), cognitive confidence (MD = 0.97, $p$ = 0.002, Cohen's

**Table 1. Sociodemographic data according to sex and circadian typology groups.**

| | Age | Sex | | | | | Circadian typology | | | | | |
|---|---|---|---|---|---|---|---|---|---|---|---|---|
| | | Men | | Women | | | Morning-type | | Neither-type | | Evening-type | |
| | Mean ± SD | n | % | n | % | $\chi^2$ | n | % | n | % | n | % | $\chi^2$ |
| **Employment status** | | | | | | | | | | | | |
| Student | 22.62 ± 3.49 | 393 | 27.1 | 242 | 29.1 | 10.81* | 78 | 14.7 | 317 | 28.4 | 240 | 37.7 | 116.43** |
| Worker | 34.68 ± 8.85 | 808 | 55.7 | 410 | 49.2 | | 370 | 69.5 | 588 | 52.7 | 260 | 40.9 | |
| Study and work | 27.81 ± 6.70 | 150 | 10.3 | 107 | 12.8 | | 61 | 11.5 | 128 | 11.5 | 68 | 10.7 | |
| No-worker | 32.31 ± 10.24 | 99 | 6.8 | 74 | 8.9 | | 23 | 4.3 | 82 | 7.4 | 68 | 10.7 | |
| **Marital status** | | | | | | | | | | | | |
| Single | 27.75 ± 8.34 | 752 | 51.9 | 353 | 42.6 | 18.18** | 191 | 36.0 | 548 | 49.2 | 366 | 57.5 | 54.04** |
| Paired | 32.85 ± 9.41 | 698 | 48.1 | 476 | 57.4 | | 339 | 64.0 | 565 | 50.8 | 270 | 42.5 | |

*$p < .05$;

**$p < .01$.

d = 0.17) and cognitive self-consciousness (MD = 0.74, $p$ = 0.02, Cohen's d = 0.39).Regarding sex, significant main effects were observed in positive beliefs about worry, need to control thoughts and cognitive self-consciousness. In this line, men scored higher than women in positive beliefs about worry (MD = 1.27, $p > .001$, Cohen's d = 0.35), need to control thoughts (MD = 1.29, $p > .001$, Cohen's d = 0.38) and cognitive self-consciousness (MD = 0.74, $p > .001$, Cohen's d = 0.23).

**Table 2. Descriptive statistics (mean ± SEM), F-tests and partial eta-square ($\eta_p^2$) for the variables.**

| | | Sex | | | | | Circadian typology | | | | | |
|---|---|---|---|---|---|---|---|---|---|---|---|---|
| | Total sample (N = 2283) | Men (n = 1450) | Women (n = 833) | F | $\eta_p^2$ | Observed power | Morning-type (n = 532) | Neither-type (n = 1115) | Evening-type (n = 636) | F | $\eta_p^2$ | Observed power |
| **Emotion regulation** | | | | | | | | | | | | |
| Cognitive reappraisal | 29.24 ± 0.14 | 28.89 ± 0.18 | 29.84 ± 0.23 | 7.01* | 0.003 | 0.75 | 30.05 ± 0.30 | 29.18 ± 0.20 | 28.66 ± 0.28 | 5.72* | 0.005 | 0.87 |
| Expressive suppression | 15.28 ± 0.12 | 16.89 ± 0.14 | 12.46 ± 0.19 | 275.45** | 0.108 | 1.00 | 14.30 ± 0.26 | 15.19 ± 0.17 | 16.25 ± 0.24 | 2.58 | 0.002 | 0.52 |
| **Metacognitions** | | | | | | | | | | | | |
| MCQ-30 total score | 62.75 ± 0.30 | 64.07 ± 0.37 | 60.44 ± 01.49 | 22.16** | 0.010 | 1.00 | 60.08 ± 0.60 | 62.52 ± 0.42 | 65.38 ± 0.58 | 6.27* | 0.005 | 0.90 |
| Positive beliefs about worry | 11.27 ± 0.09 | 11.79 ± 0.11 | 10.37 ± 0.13 | 45.42** | 0.020 | 1.00 | 10.98 ± 0.18 | 11.19 ± 0.12 | 11.66 ± 0.17 | 1.19 | 0.001 | 0.262 |
| Negative beliefs of uncontrollability and danger | 12.64 ± 0.09 | 12.61 ± 0.11 | 12.68 ± 0.15 | 0.34 | 0.000 | 0.10 | 12.14 ± 0.18 | 12.58 ± 0.13 | 13.14 ± 0.18 | 5.65* | 0.005 | 0.86 |
| Cognitive confidence | 11.16 ± 0.09 | 11.21 ± 0.12 | 11.35 ±0.16 | 0.31 | 0.000 | 0.09 | 10.81 ± 0.17 | 11.31 ± 0.13 | 11.56 ± 0.19 | 6.38* | 0.006 | 0.90 |
| Need to control thoughts | 11.49± 0.08 | 12.00 ± 0.10 | 10.6 ± 0.12 | 57.38** | 0.025 | 1.00 | 10.85 ± 0.16 | 11.47 ± 0.11 | 12.05 ± 0.15 | 2.45 | 0.002 | 0.50 |
| Cognitive self-consciousness | 16.09 ± 0.09 | 16.46 ± 0.12 | 15.45 ± 0.15 | 13.89** | 0.006 | 0.96 | 15.29 ± 0.19 | 15.97 ± 0.13 | 16.97 ± 0.17 | 4.79* | 0.004 | 0.80 |
| **Assertiveness** | 0.27 ± 0.58 | 1.13 ± 0.71 | -1.21 ± 0.98 | 3.80 | 0.002 | 0.50 | 4.25 ± 1.27 | -0.66 ± 0.79 | -1.41 ± 1.10 | 5.73* | 0.005 | 0.87 |

*$p < .01$;

**$p < .001$.

## Assertiveness

Table 2 shows descriptive data structured for the total sample, by sex and by circadian typology groups in the Rathus Assertiveness Schedule. A significant main effect was observed only for circadian typology. Post-hoc comparisons showed higher scores in morning-type participants as compared to evening- (MD = 4.78, $p$ = 0.02, Cohen's $d$ = 0.20) and to neither-type (MD = 4.86, $p$ = 0.004, Cohen's $d$ = 0.18).

## Regression analyses

Table 3 shows results of multiple regression analyses while Tables 4 and 5 show the model coefficients for each variable. The analyses revealed that sex and age were significantly related to cognitive reappraisal (explaining a 0.4% of the variance), expressive suppression (14.2%), assertiveness (0.7%), metacognitions (3.9%), positive beliefs about worry (5.2%), need to control thoughts (5.5%) and cognitive self-consciousness (6.6%), with $p$ < .05 in all cases. Moreover, the inclusion of the rMEQ scores in the equation resulted in a significantly increase of the explained variance for cognitive reappraisal (0.5%), expressive suppression (0.3%), assertiveness (0.6%), metacognitions (0.8%), negative beliefs of uncontrollability and danger (0.9%), cognitive confidence (0.7%), need to control thoughts (0.3%) and cognitive self-consciousness (0.5%), with $p$ < .05 in all cases.

## Discussion

In this study, and for the first time, the relationships among circadian typology, emotion regulation strategies, metacognitions and assertiveness were examined in a wide sample of healthy subjects. Participants distribution according to the morningness-eveningness dimension was skewed toward eveningness, which is in line with previous studies performed with large samples [3,4].

Obtained results supported the hypothesis that morning-type participants showed the highest cognitive reappraisal, meaning a higher tendency to use this kind of antecedent-focused emotion regulation strategy, which implies the reappraise of a potentially emotion-eliciting situation in order to modify its meaning and its emotional impact. Moreover, in line with the regression results, it was found that the closer one is to the morningness pole, the greater tendency to use this kind of emotion regulation strategy and the lower to perform a response-focused emotion regulation strategy as expressive suppression is, and vice versa. According with previous studies, men showed a greater tendency to suppress their emotional expression [32,33]. Moreover, women tendency to reappraise potentially emotion-eliciting situations was higher than men, which constitutes a difference regarding previous works [32,33]. Nevertheless, the absence of interactive effect between sex and circadian typology suggests that the circadian typology obtained results are independent of sex.

This higher tendency to cognitive reappraise observed in morning-type fits in with those studies that state this chronotype as a protective factor against psychological problems [6,8,11], as the usage of this emotion regulation strategy together with morningness tendency has been linked to well-being [32,67] and quality of life [68–70], as well as to several variables that acts as protective factor against psychopathology such as positive affect [71,72], psychological adjustment [32,73] and resilience [3,74], among others.

Moreover, regarding regression results, a higher use of expressive suppression was observed for those participants closer to the eveningness pole. Expressive suppression is a well-known maladaptive strategy that is related with diverse psychological problems such as negative affect and depression [32,75,76], as well as with several factors related to those like lower interpersonal functioning, well-being, life satisfaction, self-esteem, optimism and emotional

**Table 3. Results of stepwise multiple regression analyses.**

| | First step | | | Second step | | |
| | Sex & age | | | Sex, age & rMEQ scores | | |
| | $R^2$ | $F_{(2,2280)}$ | $f^2$ | $R^2$ | $F_{(3,2279)}$ | $f^2$ |
|---|---|---|---|---|---|---|
| **Emotion regulation** | | | | | | |
| Cognitive reappraisal | .004 | 5.60* | .004 | .009 | 7.90** | .009 |
| Expressive suppression | .142 | 189.55** | .165 | .144 | 129.33** | .168 |
| **Assertiveness** | .007 | 9.18** | .007 | .012 | 10.39** | .012 |
| **Metacognitions** | | | | | | |
| MCQ-30 total score | .039 | 47.29** | .040 | .047 | 38.10** | .049 |
| Positive beliefs about worry | .052 | 63.36** | .054 | .051 | 42.23** | .054 |
| Negative beliefs of uncontrollability and danger | .001 | 2.37 | .001 | .010 | 8.76** | .010 |
| Cognitive confidence | .001 | 2.68 | .001 | .008 | 6.78** | .008 |
| Need to control thoughts | .055 | 67.48** | .058 | .057 | 47.31** | .060 |
| Self-consciousness | .066 | 81.11** | .071 | .070 | 58.46** | .075 |

$^*p < .01$,

$^{**}p < .001$.

intelligence [32]. In this line, the higher use of maladaptive emotion regulation strategies, like expressive suppression, and the lower use of adaptive ones, as cognitive reappraisal, observed for those people closer to the eveningness pole reinforces the assumption of the evening-type as a risk factor for the development of psychological problems and pathologies.

Regarding metacognitions, the hypothesis that evening-type participants show the highest tendency to suffer maladaptive cognitions about the thinking is supported by the results, while neither-type subjects tendency is between both extreme groups. Concretely, it was observed that evening-type subjects showed the highest tendency to show negative beliefs about uncontrollability and danger ("worry is uncontrollable" or "my worry is dangerous for me"), to distrust own cognitive memory ("I do not trust my memory" or "my memory can mislead me at times"), and to monitor the attention to one's thoughts ("I think a lot about my thoughts" or "I am constantly aware of my thinking"). In this line, and according to the regression results, the closer one is to the eveningness pole, the higher the tendency is to suffer more maladaptive metacognitions, specifically worries about worrying, distrust own memory, monitor thoughts, and feel the need to control thoughts. Moreover, in line with the results of the Spanish adaptation of the MCQ-30 [62] men showed the highest tendency to suffer maladaptive thinking about own cognitions such as positive beliefs about worry ("worry helps me to solve problems or to work well") and need to control thoughts ("it is bad to think certain thoughts" or "not

**Table 4. Multiple linear regression model coefficients for emotion regulation strategies, assertiveness and MCQ-30 total score.**

| | Cognitive reappraisal | | | Expressive suppression | | | Assertiveness | | | Metacognitions | | |
| | B | SE B | β | B | SE B | β | B | SE B | β | B | SE B | β |
|---|---|---|---|---|---|---|---|---|---|---|---|---|
| **Step 1** | | | | | | | | | | | | |
| Sex | -0.013 | 0.015 | -0.017 | -0.071 | 0.012 | -0.112 | 0.239 | 0.063 | 0.080 | -0.247 | 0.032 | -0.159 |
| Age | 0.989 | 0.297 | 0.070 | -4.256 | 0.240 | -0.346 | -2.934 | 1.205 | -0.051 | -3.018 | 0.617 | -0.101 |
| **Step 2** | | | | | | | | | | | | |
| Sex | -0.029 | 0.016 | -0.040 | -0.061 | 0.013 | -0.095 | 0.170 | 0.065 | 0.057 | -0.204 | 0.033 | -0.132 |
| Age | 0.897 | 0.298 | 0.064 | -4.198 | 0.241 | -0.342 | -3.310 | 1.206 | -0.058 | -2.783 | 0.617 | -0.093 |
| rMEQ | 0.121 | 0.034 | 0.077 | -0.077 | 0.028 | -0.057 | 0.495 | 0.139 | 0.078 | -0.309 | 0.071 | -0.094 |

**Table 5. Multiple linear regression model coefficients for the different metacognitions assessed through MCQ-30.**

| | Positive beliefs about worry | | | Negative beliefs of uncontrollability and danger | | | Cognitive confidence | | | Need to control thoughts | | | Self-consciousness | | |
|---|---|---|---|---|---|---|---|---|---|---|---|---|---|---|---|
| | B | SE B | β | B | SE B | β | B | SE B | β | B | SE B | β | B | SE B | β |
| **Step 1** | | | | | | | | | | | | | | | |
| Sex | -0.073 | 0.009 | -0.162 | -0.021 | 0.010 | -0.045 | 0.022 | 0.010 | 0.047 | -0.063 | 0.008 | -0.155 | -0.113 | 0.010 | -0.235 |
| Age | -1.238 | 0.178 | -0.143 | 0.123 | 0.187 | 0.014 | 0.077 | 0.195 | 0.008 | -1.247 | 0.160 | -0.160 | -0.733 | 0.188 | -0.080 |
| **Step 2** | | | | | | | | | | | | | | | |
| Sex | -0.073 | 0.010 | -0.162 | -0.007 | 0.010 | -0.015 | 0.034 | 0.011 | 0.072 | -0.056 | 0.009 | -0.138 | -0.102 | 0.010 | -0.213 |
| Age | -1.239 | 0.179 | -0.143 | 0.199 | 0.187 | 0.023 | 0.143 | 0.195 | 0.015 | -1.210 | 0.161 | -0.155 | -0.675 | 0.188 | -0.073 |
| rMEQ | 0.001 | 0.021 | 0.001 | -0.100 | 0.021 | -0.102 | -0.087 | 0.022 | -0.085 | -0.048 | 0.018 | -0.055 | -0.076 | 0.022 | -0.075 |

being able to control my thoughts is a sign of weakness"), as well as the lowest cognitive confidence. Nevertheless, the non-interaction observed between sex and circadian typology indicates that circadian typology results are independent of sex.

The higher tendency to hold negative beliefs of uncontrollability and danger, to distrust the own memory and to stay continuously aware of self-thinking observed in evening-type supports the assumption of this typology as a risk factor for the development of different psychological problems and pathologies [9,11], as these metacognitive beliefs are related to diverse problems such as drug consumption and addictive behaviors [47,77,78], anxiety and stress [45,46,79] and depressive symptomatology [41,80,81], among others.

The assertiveness results support the hypothesis of a greater tendency to affirm a right or a point of view without either aggressively threatening the rights of another nor submissively allow another to deny one's rights or point of view in the morning-type, which contrasts with the lower ability in the evening-type, while neither-type individuals are in an intermediate position. Likewise, according to the regression results, it was also found that the morningness-eveningness dimension was directly related with this ability. Moreover, and in line with previous results [82], the absence of sex differences for this skill suggests that obtained results are independent of sex.

Assertiveness constitutes a core skill for well-being which is positively implied in happiness [58], self-esteem [54] and relationship satisfaction [56], as well as in the amelioration of diverse psychological disturbances [51]. Thus, the higher capacity to affirm a right respectfully observed in the morning-type is in line with the consideration of this circadian typology as a protective factor against the development of psychological issues. Likewise, low assertiveness has been found related with psychological problems like depression [52], anxiety [50,53] and major mental problems [55], issues that are more prone in evening-type, reinforcing the consideration of this chronotype as a risk factor for those.

Altogether, although there was some sex differences observed for the emotion regulation strategies and metacognitions, the absence of interactive effects between sex and circadian typology suggests that obtained results for emotion regulation strategies, metacognitions and assertiveness results could provide additional evidence for a better understanding of the relationship between circadian typology and psychological characteristics which might underlie their associations with psychological problem and strengths. Likewise, it must be noted that differences found between evening and morning-type participants remain significant even once age and sex are removed from the model. Results might be explained by the social jet lag theory [16], which hypothesizes that the social jet-lag sufferers, mainly evening-type, must set up strategies in order to adapt to or to mitigate the misalignment between their biological and

the social clock, which is commonly oriented towards morningness. In this line, evening-type persons, who are the mainly social jet-lag sufferers, must perform different functions (sleep and wake-up, eat, work, study, child care, etc.) in a schedule that is unadjusted to their biological clock, which might be understood as upstream swim, that might result in higher fatigue as it has been observed in previous works [83–85]. In this line, longitudinal studies are needed to analyze if this worse emotion regulation strategies together with the maladaptive metacognitive style and the lower assertiveness observed in those participants closer to the eveningness pole might be the result of this higher fatigue.

This study is not exempt of limitations. One of them is the low control of the on-line data collection as compared with traditional paper questionnaires. Moreover, the presence of psychological and medical problems, which was assessed by self-report questions and not by physical and psychological or psychiatric interviews constitute a weakness. Age and sex proportions of the sample are also limitations. In this line, there was a wide proportion of participants aged 20–30, which differs from Spanish population normative data and that might be attributed to the way the study was spread (university and social networks mainly). Likewise, the higher proportion of men can be also considered as a weakness. Finally, regression results, although significant, are relatively low, implying that conclusions derived from those should be made cautiously.

Summarizing, this is the first study which assesses the associations between circadian typology, emotion regulation strategies, metacognitions and assertiveness in a wide sample of healthy adults. Morning-type participants showed the higher use of cognitive reappraisal, the higher assertiveness and the lower amount maladaptive metacognitions while evening-type showed the lower use of cognitive reappraisal, the lower assertiveness and the higher amount maladaptive metacognitions, while the neither-type participants hold an intermediate position between the extreme groups. The results emphasize that circadian typology is related to psychological well-being in healthy population, highlighting the assumption of the morning-type as a protective factor against the development of psychological issues as well as the consideration of the evening-type as a risk factor for the development of diverse problems like depression, anxiety and substance use. Likewise, obtained results may improve the understanding of the associations between circadian typology and different psychological factors related to well-being and psychological health and psychopathology by offering possible explanations of their relationships. In addition, the results may become useful for healthcare professional, mostly psychologists, that should take into account the circadian typology and the social jet-lag when developing meta-cognitive therapy and psychotherapeutic processes orientated to improving the emotion regulation strategies and assertiveness, as well as for the development of prevention and health promotion programs. Further research, mainly longitudinal, is needed for a better understanding of the relationships between circadian typology and emotion regulation, metacognitions and assertiveness.

## Acknowledgments

I wish to thank Yolanda Casado for her support and feedback during the study design, data collection and writing, Fernando Antúnez for him support during the writing, and all participants as this study would have not been ever possible without them.

## Author Contributions

**Conceptualization:** Juan Manuel Antúnez.

**Data curation:** Juan Manuel Antúnez.

**Formal analysis:** Juan Manuel Antúnez.

**Investigation:** Juan Manuel Antúnez.

**Methodology:** Juan Manuel Antúnez.

**Writing – original draft:** Juan Manuel Antúnez.

**Writing – review & editing:** Juan Manuel Antúnez.

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
