## [Decision Letter · Decision Letter 0]

30 Oct 2019

PONE-D-19-23969

Circadian typology is related to emotion regulation, metacognitive beliefs and assertiveness in healthy adults

PLOS ONE

Dear Dr. Antúnez,

Thank you for submitting your manuscript to PLOS ONE. After careful consideration, we feel that it has merit but does not fully meet PLOS ONE’s publication criteria as it currently stands. Therefore, we invite you to submit a revised version of the manuscript that addresses the points raised during the review process.

I have been able to secure useful feedback from one Reviewer. Please, see the specific comments at the bottom of this letter. I would not repeat these comments here. This evaluation of your manuscript highlights that there were several major concerns with the current version of the study. These concerns could be addressed in a reviewed version of your study.

However, because this may be considered as a major review, please notice that a resubmission will require another round of reviews, hopefully from more that just one Reviewer, and that the final outcome of this process cannot be predicted at this point. If you decide to resubmit, please address each of the concerns that were reaised in the current review.

We would appreciate receiving your revised manuscript by Dec 13 2019 11:59PM. To enhance the reproducibility of your results, we recommend that if applicable you deposit your laboratory protocols in protocols.io, where a protocol can be assigned its own identifier (DOI) such that it can be cited independently in the future. For instructions see: http://journals.plos.org/plosone/s/submission-guidelines#loc-laboratory-protocols

We look forward to receiving your revised manuscript.

Kind regards,

Angel Blanch, Ph.D.

Academic Editor

PLOS ONE

Journal Requirements:

1. 

https://www.tandfonline.com/doi/abs/10.3109/07420528.2015.1008700?journalCode=icbi20

In your revision ensure you cite all your sources (including your own works), and quote or rephrase any duplicated text outside the methods section. Further consideration is dependent on these concerns being addressed.

Reviewers' comments:

Reviewer's Responses to Questions

**Comments to the Author**

1. Is the manuscript technically sound, and do the data support the conclusions?

Reviewer #1: Yes

2. Has the statistical analysis been performed appropriately and rigorously? 

Reviewer #1: I Don't Know

3. Have the authors made all data underlying the findings in their manuscript fully available?

Reviewer #1: Yes

4. Is the manuscript presented in an intelligible fashion and written in standard English?

Reviewer #1: No

5. Review Comments to the Author

Reviewer #1: PONE-D-19-02089

Title: Circadian typology is related to emotion regulation, metacognitive beliefs and

assertiveness in healthy adults

In his manuscript entitled “Circadian typology is related to emotion regulation, metacognitive beliefs and assertiveness in healthy adults”, J.M. Antúnez addresses potential associations between self-reported circadian preference (evening vs. morning types) and several other personality-related traits (as indexed by questionnaire measures), providing partial support for the hypothesis of higher prevalence of dysfunctional beliefs/strategies in individuals reporting a subjective bias toward later circadian phases.

Overall, this is an interesting paper based on a large-sample study with considerable relevance to the field. However, I have several concerns with respect to some methodological issues as well as the interpretations of findings and final conclusions drawn by the author (as outlined in more detail below).

Moreover, throughout the manuscript (starting with the first sentence) there are many instances of awkward language, regarding both non-idiomatic style (i.e., choice of words) and grammatical errors (mostly minor aberrations, such as lack of 3rd person “-s”, e.g., yet sometimes more severe cases, such as incomplete sentences). The text should therefore be thoroughly proofread by a native speaker before final acceptance for publication can be considered.

In a similar vein, use of references styles is not entirely consistent across the manuscript (e.g., full author references on p. 6).

Apart from some problems with grammar and phrasing (see above), both Abstract and Introduction are well-written and clear in stating the present study’s background and main objectives.

The description of the methods is straightforward and clear. (Minor issue: There seems to be a misplaced decimal marker in the description of the internal consistency of the ERQ scales.)

The Results section is also largely consistent. However, it is unclear how exactly post-hoc testing was performed: What do the numbers in brackets indicate (t-values, F-values …)? (Minor issue: There is probably a wrong “>” symbol on p. 10.)

Moreover: How do the results look like when age and sex are not entered as predictors (in ANCOVA or multiple regression, respectively)? Are the differences between evening and morning type still significant? Most importantly: Is the increase in explained variance (incremental variance) by inclusion of the rMEQ scores, as described on pp. 13-14, significant? The author should include this important information and also consider reporting all respective regression weights for each model.

If the finding of an association of circadian typology and other traits depends on statistical control for age and sex (as I am inclined to suspect on the basis of the information provided), this should be highlighted and clearly addressed as a limitation in the Discussion. Also, given that the results are based purely on self-report as well as a correlational design without longitudinal assessment, I would strongly recommend phrasing the conclusions in a much more cautious way, since neither is there direct evidence for the alleged association between psychopathology and circadian typology (discussed as a “risk factor” by the author) nor is there any indication of a direct causal link between circadian typology and emotion regulation (i.e., the design of the study is not at all appropriate for elucidating the direction of causation at all).

Taken together, most of my concerns may be addressed in a thorough revision of the manuscript. Therefore, this could become an interesting and valuable contribution to the literature, given that the manuscript would have been proofread and corrected with respect to grammar and style, the conclusions would be expressed in a more cautious way, and the shortcomings of the study (as described above) would be explicitly addressed.

6. PLOS authors have the option to publish the peer review history of their article (what does this mean?). If published, this will include your full peer review and any attached files.

Reviewer #1: No

---

## [Author Response · Author response to Decision Letter 0]

28 Nov 2019

Dear Prof Blanch,

Thank you for giving me the opportunity to revise my work for your further consideration. I have considered all the points that have been required by the journal and suggested by the reviewer which has allowed me to improve the manuscript. Next, I reply to each of the comments in a detailed and itemized form. 

1. Please ensure that your manuscript meets PLOS ONE’s style requirements, including those for file naming.

Attending to your requirement, manuscript meets PLOS ONE’s style requirements.

2. We noticed you have some minor occurrence of overlapping text with the following previous publication, which needs to be addressed: https://www.tandfondline.com/doi/abs/10.3109/07420528.215.1008700?journalCode=icbi20 In your revision ensure you cite all your sources (including your own works), and quote or rephrase any duplicated text outside the methods section.

Regarding your requirement, minor overlapping text with other previous publications has been edited.

Reviewer comments:

1. Moreover, throughout the manuscript (starting with the first sentence) there are many instances of awkward language, regarding both non-idiomatic style (i.e., choice of words) and grammatical errors (mostly minor aberrations, such as lack of 3rd person “-s”, e.g., yet sometimes more severe cases, such as incomplete sentences). The text should therefore be thoroughly proofread by a native speaker before final acceptance for publication can be considered.

Regarding your suggestion, the manuscript has been proofread by a native speaker.

2. In a similar vein, use of references styles is not entirely consistent across the manuscript (e.g., full author references on p. 6).

Attending to your comment, all references have been revised to ensure the style consistency across the manuscript.

3. There seems to be a misplaced decimal marker in the description of the internal consistency of the ERQ scales.

Regarding your comment, misplaced decimal marker in the description of the internal consistency of the ERQ scales has been corrected.

4. The Results section is also largely consistent. However, it is unclear how exactly post-hoc testing was performed: What do the numbers in brackets indicate (t-values, F-values …)? 

Attending to your recommendation, brackets that indicate mean differences (MD) have been modified so every reader will understand its meaning.

5. (Minor issue: There is probably a wrong “>” symbol on p. 10.)

Regarding your comment, the wrong “>” symbol on p. 10 has been modified to “<”. 

6. Moreover: How do the results look like when age and sex are not entered as predictors (in ANCOVA or multiple regression, respectively)? Are the differences between evening and morning type still significant? Most importantly: Is the increase in explained variance (incremental variance) by inclusion of the rMEQ scores, as described on pp. 13-14, significant? The author should include this important information and also consider reporting all respective regression weights for each model.

ANCOVA and MANCOVA analyses without age and sex as independent and covariate respectively have a considerable impact in the results as F-scores for every variable, except for cognitive confidence, were increased (even positive beliefs difference between morning and evening-types become significant with this change). In the case of cognitive confidence, F-score decreased but group differences remained significant.

Regarding regression analyses, the non-inclusion of age and sex as predictors showed significant results for every variable: Cognitive reappraisal (R2 = .005, F = 12.21, p < .001), Expressive suppression (R2 = .016, F = 37.90, p < .001), assertiveness (R2 = .007, F = 18.08, p < .001), MCQ-30 total score (R2 = .021, F = 49.45, p < .001), positive beliefs about worry (R2 = .004, F = 10.03, p = .002), negative beliefs of uncontrollability and danger (R2 = .010, F = 24.81, p < .001), cognitive confidence (R2 = .003, F = 8.56, p = .003), need to control thoughts (R2 = .013, F = 31.44, p < .001) and self-consciousness (R2 = .022, F = 51.84, p < .001).

Moreover, attending to your recommendation, the increase in explained variance significance has been included in the manuscript. 

Age and sex were included as factor and covariate respectively to make circadian typology results more accurate. That is because circadian typology is clearly related to age while sex has been found related to diverse variables such as suppress expression, and these relations might affect the results. The aim of the study was to analyze the relationship between circadian typology and metacognitions, emotion regulation and assertiveness, by controlling and neutralizing two factors (age and sex) that may underlie several associations and boost our results unless those were controlled.

Regarding your recommendation, regression coefficients (B, SE B and β) for each model were reported in Tables 4 and 5.

7. If the finding of an association of circadian typology and other traits depends on statistical control for age and sex (as I am inclined to suspect on the basis of the information provided), this should be highlighted and clearly addressed as a limitation in the Discussion. 

Associations between circadian typology and other traits did not depend on statistical control for age and sex. Moreover, the exclusion of sex and age as controlled variables resulted in higher weight on the associations.

8. Also, given that the results are based purely on self-report as well as a correlational design without longitudinal assessment, I would strongly recommend phrasing the conclusions in a much more cautious way, since neither is there direct evidence for the alleged association between psychopathology and circadian typology (discussed as a “risk factor” by the author) nor is there any indication of a direct causal link between circadian typology and emotion regulation (i.e., the design of the study is not at all appropriate for elucidating the direction of causation at all).

Attending to your recommendation, conclusions have been re-phrased in a much more cautious way, avoiding any term that may suggest any direct causal link between circadian typology and any one of the studied traits nor direct evidence for the alleged association between psychopathology and circadian typology:

“Altogether, although there was some sex differences observed for the emotion regulation strategies and metacognitions, the absence of interactive effects between sex and circadian typology suggests that obtained results for emotion regulation strategies, metacognitions and assertiveness results could provide additional evidence for a better understanding of the relationship between circadian typology and psychological characteristics which might underlie their associations with psychological problem and strengths. Results might be explained by the social jet lag theory [16], which hypothesizes that the social jet-lag sufferers, mainly evening-type, must set up strategies in order to adapt to or to mitigate the misalignment between their biological and the social clock, which is commonly oriented towards morningness. In this line, evening-type persons, who are the mainly social jet-lag sufferers, must perform different functions (sleep and wake-up, eat, work, study, child care, etc.) in a schedule that is unadjusted to their biological clock, which might be understood as upstream swim, leading to a widespread fatigue for cognitive, emotional and behavioral systems. In this line, longitudinal studies are needed to analyze if this worse emotion regulation strategies together with the maladaptive metacognitive style and the lower assertiveness observed in those participants closer to the eveningness pole might be the result of this higher fatigue.”

---

## [Decision Letter · Decision Letter 1]

3 Feb 2020

PONE-D-19-23969R1

Circadian typology is related to emotion regulation, metacognitive beliefs and assertiveness in healthy adults

PLOS ONE

Dear Dr. Antúnez,

Thank you for submitting your manuscript to PLOS ONE. After careful consideration, we feel that it has merit but does not fully meet PLOS ONE’s publication criteria as it currently stands. Therefore, we invite you to submit a revised version of the manuscript that addresses the points raised during the review process.

This version of the manuscript has been evaluated by Reviewer #1, who provided initial feedback about your study in the previous round of reviews. In addition, there are additional comments from a new Reviewer (#2). Please, see their comments at the bottom of this letter. As you will see, there is still room for further improving the presentation of your study. Therefore, the manuscript should be reviewed again in accordance with these latter suggestions.  

We would appreciate receiving your revised manuscript by Mar 19 2020 11:59PM. To enhance the reproducibility of your results, we recommend that if applicable you deposit your laboratory protocols in protocols.io, where a protocol can be assigned its own identifier (DOI) such that it can be cited independently in the future. For instructions see: http://journals.plos.org/plosone/s/submission-guidelines#loc-laboratory-protocols

We look forward to receiving your revised manuscript.

Kind regards,

Angel Blanch, Ph.D.

Academic Editor

PLOS ONE

Reviewers' comments:

Reviewer's Responses to Questions

**Comments to the Author**

1. If the authors have adequately addressed your comments raised in a previous round of review and you feel that this manuscript is now acceptable for publication, you may indicate that here to bypass the “Comments to the Author” section, enter your conflict of interest statement in the “Confidential to Editor” section, and submit your "Accept" recommendation.

Reviewer #1: (No Response)

Reviewer #2: (No Response)

2. Is the manuscript technically sound, and do the data support the conclusions?

Reviewer #1: Yes

Reviewer #2: Partly

3. Has the statistical analysis been performed appropriately and rigorously? 

Reviewer #1: Yes

Reviewer #2: Yes

4. Have the authors made all data underlying the findings in their manuscript fully available?

Reviewer #1: Yes

Reviewer #2: Yes

5. Is the manuscript presented in an intelligible fashion and written in standard English?

Reviewer #1: Yes

Reviewer #2: No

6. Review Comments to the Author

Reviewer #1: After a second, thorough reading I conclude that all of my concerns have been sufficiently addressed and the quality of the paper has improved considerably.

I appreciate the author’s methodological scrutiny in controlling for potential confounding variables. (i.e., age and sex). However, the author should add a sentence or two (e.g., in the Discussion) asserting to the reader that the differences between evening and morning type are still significant even if age and sex are left out of the model.

Minor: Typo “lineal regression” in headings of Tables 4 and 5.

Reviewer #2: The author presents the results of an online study about the link between morningness-eveningness (circadian typology) and adaptive or maladaptive habits in emotion regulation, metacognitive beliefs and assertiveness. I think the manuscript is well suited for publication in PLOSone but still needs some adjustments.

Major aspects:

“Likewise, significant differences were observed between women (14.78 ± 0.15) and men (13.67 ± 0.11) in rMEQ scores (t(1,2281) = 5.89, p < 0.001, Cohen’s d = 0.25).” Does this mean that more women are morning types?

Related to that how does this match with the statements: “Nevertheless, the absence of interactive effect between sex and circadian typology suggests that the circadian typology obtained results are independent of sex.”and “Nevertheless, the non-interaction observed between sex and circadian typology indicates that circadian typology results are independent of sex”

Isn’t the interesting aspect here that morningness-eveningness can contribute to the explanation of emotion regulation, metacognitive beliefs and assertiveness beyond sex as revealed by the regression analysis?

The author writes “Participants distribution according to the morningness- eveningness dimension was skewed toward eveningness, which is in line with previous studies performed with large samples [3,4].” Does this mean that we are facing a representative sample here? Further comparisons might be useful. What about the mentioned usual distribution of 20/60/20 % of morning/indifferent/evening types – is this found within the sample?

When fatigue is discussed I was wondering whether there is any evidence yet that supports this idea, that evening-types suffer from fatigue? If I got the reasoning right that is deduced from social jet-lag theory evening-types (working against their biological clock) that have to adapt to a more morning-oriented social clock should experience greater fatigue which brings up maladaptive cognitive, emotional and behavioral habits. What about other groups of persons or cultures?

Minor aspects:

I think the english language still needs some corrections e.g. “…non-found…”; “…the hypothesis that evening-type participants show(s)…”; “One of them is the lower absence of control of the on-line data collection…”

I was wondering how many participants were excluded due to duplicate answers and how many because they did not meet the inclusion criteria. Also the inclusion/exclusion criteria should be named half a page earlier when the exclusion is mentioned. Does inclusion/exclusion significantly change the results of the present study?

Please explain the score range for the rMEQ. What are the scales?

Decimal separators are inkonsistent in the participants section and in Table 1.

“…and evening-type (27.49 ± 0.30 yrs; p < 0.001), who were older than neither-type (p < 0.001).” I believe it should read younger, if the the numbers are in brackets are means.

The metacognitions sections would be easier to follow when all post-hoc comparisons would follow the description of significance directly.

I think the section regression analysis is not easy to follow linguistically as links are not clear. Also: was the increase from the first to the second model significant?

Table 2 should be ordered the same way as the results section.

Table 4 und 5 are named alike. Aren’t the information in Table 5 subscales of metacognitive beliefs?

Please provide english explanations for the variables in the .sav-file on OSF.

7. PLOS authors have the option to publish the peer review history of their article (what does this mean?). If published, this will include your full peer review and any attached files.

Reviewer #1: No

Reviewer #2: No

---

## [Author Response · Author response to Decision Letter 1]

12 Feb 2020

Reviewer #1:

1. The author should add a sentence or two (e.g., in the Discussion) asserting to the reader that the differences between evening and morning type are still significant even if age and sex are left out of the model.

Regarding your suggestion, the assertion that the differences between evening- and morning-type are still significant even if age and sex are left out of the model has been added to the discussion: 

“Likewise, it must be noted that differences found between evening and morning-type participants remain significant even once age and sex are removed from the model.”

2. Minor: Typo “lineal regression” in headings of Tables 4 and 5.

According to your correction, “lineal regression” has been modified to “linear regression” in both headings of Tables 4 and 5.

Reviewer #2: 

1. “Likewise, significant differences were observed between women (14.78 ± 0.15) and men (13.67 ± 0.11) in rMEQ scores (t(1,2281) = 5.89, p < 0.001, Cohen’s d = 0.25).” Does this mean that more women are morning types? 

Not really, it is needed a score between 18 and 25 in rMEQ to be considered as a morning-type. Nevertheless, it can be said that women are more morning-type than men.

2. Related to that how does this match with the statements: “Nevertheless, the absence of interactive effect between sex and circadian typology suggests that the circadian typology obtained results are independent of sex.”and “Nevertheless, the non-interaction observed between sex and circadian typology indicates that circadian typology results are independent of sex”. Isn’t the interesting aspect here that morningness-eveningness can contribute to the explanation of emotion regulation, metacognitive beliefs and assertiveness beyond sex as revealed by the regression analysis? 

Absolutely. Both statements are not incompatible with the greater tendency in women toward morningness. Despite this result, no interactive effect between sex and circadian typology in ANCOVA and MANCOVA analyses was found for any of the assessed variables. In other words, although sex exerted an important influence in some variables, carried out analyses revealed that circadian typology was also clearly related to the studied variables regardless of sex, as it has been shown in regression analyses.

3. The author writes “Participants distribution according to the morningness- eveningness dimension was skewed toward eveningness, which is in line with previous studies performed with large samples [3,4].” Does this mean that we are facing a representative sample here? Further comparisons might be useful. What about the mentioned usual distribution of 20/60/20 % of morning/indifferent/evening types – is this found within the sample? 

The present sample comprised a 27.9% of evening-type, 48.8% of neither-type, and 23.3% of morning-type participants. I assume that the 27.9% rate of evening-type participants might be due to the wide proportion of participants aged 20-30, that differs from the European population (where the wider proportion rounds 35-60 years according to eurostat).

4. When fatigue is discussed I was wondering whether there is any evidence yet that supports this idea, that evening-types suffer from fatigue? 

Different studies have observed a higher fatigue among evening-type (Fárková, Šmotek, Bendová, Manková, & Kopřivová, 2019; Furusawa et al., 2015; Jeon, Bang, Park, Kim, & Yoon, 2017) . Taking your comment into account, those references have been added to the text. Moreover, as this study did not analyze the fatigue and cause-effect, the fatigue-related phrase has been modified:

“In this line, evening-type persons, who are the mainly social jet-lag sufferers, must perform different functions (sleep and wake-up, eat, work, study, child care, etc.) in a schedule that is unadjusted to their biological clock, which might be understood as upstream swim, that might result in higher fatigue as it has been observed in previous works [83–85].”

5. If I got the reasoning right that is deduced from social jet-lag theory evening-types (working against their biological clock) that have to adapt to a more morning-oriented social clock should experience greater fatigue which brings up maladaptive cognitive, emotional and behavioral habits. What about other groups of persons or cultures?

It will always depends on the synchrony between the social (cultural) clock and the biological clock of any individual.

6. I think the english language still needs some corrections e.g. “…non-found…”; “…the hypothesis that evening-type participants show(s)…”; “One of them is the lower absence of control of the on-line data collection…

Attending to your comment, corrections have been performed.

7. I was wondering how many participants were excluded due to duplicate answers and how many because they did not meet the inclusion criteria. 

Fifty-one duplicate answers were found. First, data was introduced into a Microsoft Excel sheet and a function that highlights all duplicate rows was performed. Second, the absence of very similar responses was checked manually. For this purpose, I checked responses that were sent almost simultaneously. Finally, email duplicates were looked for. For those, the similarity of the replies to the different questionnaires, specially to sociodemographic data, was checked. In the event that sociodemographic data were different, this was not considered as a duplicate answer.

8. Also the inclusion/exclusion criteria should be named half a page earlier when the exclusion is mentioned. 

According to your recommendation, inclusion/exclusion criteria has been named following the exclusion mention: 

“The total amount of participants in the study was 3016, out of which 733 were excluded due to duplicate answers and for non-accomplishing the inclusion criteria (Spanish resident, absence of mental and physical health problems, absence of shift work and 18-60 yrs).”

9. Does inclusion/exclusion significantly change the results of the present study?

Yes. If exclusion criteria are not applied the obtained results are stronger in terms of new significant associations as well as in term of statistical results (p, R2, F, partial eta square and observed power). 

10. Please explain the score range for the rMEQ. What are the scales?

rMEQ is a 25-items questionnaire that can be scored between 4 and 25. The lower the scores are, the closer one is to the eveningness pole; The higher the scores are, the closer one is to the morningness pole. Moreover, this questionnaire allows to classify participants into three circadian typology categories: evening-type (for scores between 4 and 11), neither-type (for scores between 12 and 17), and morning-type (for scores between 18 and 25).

11. Decimal separators are inkonsistent in the participants section and in Table 1.

Attending to your comment, decimals separators in participants section and in Table 1 have been corrected.

12. “…and evening-type (27.49 ± 0.30 yrs; p < 0.001), who were older than neither-type (p < 0.001).” I believe it should read younger, if the the numbers are in brackets are means.

Attending to your correction, older has been changed to younger.

13. The metacognitions sections would be easier to follow when all post-hoc comparisons would follow the description of significance directly.

Attending to your recommendation, metacognitions section has been modified as follows:

“Significant main effects were observed in MCQ-30 total score, negative beliefs of uncontrollability and danger, cognitive confidence and cognitive self-consciousness for circadian typology. According to post-hoc comparisons, neither-type participants showed higher scores than morning-type in cognitive confidence (MD = 0.70, p = 0.012, Cohen’s d = 0.12) and scored lower than evening-type in MCQ-30 total score (MD = 1.94, p = 0.032, Cohen’s d = 0.21) and cognitive self-consciousness (MD = 0.65, p = 0.016, Cohen’s d = 0.23). Evening-type subjects showed higher scores than morning-type in MCQ-30 total score (MD = 3.13, p = 0.002, Cohen’s d = 0.17), negative beliefs of uncontrollability and danger (MD = 0.91, p = 0.002, Cohen’s d = 0.23), cognitive confidence (MD = 0.97, p = 0.002, Cohen’s d = 0.17) and cognitive self-consciousness (MD = 0.74, p = 0.02, Cohen’s d = 0.39).Regarding sex, significant main effects were observed in positive beliefs about worry, need to control thoughts and cognitive self-consciousness. In this line, men scored higher than women in positive beliefs about worry (MD = 1.27, p > .001, Cohen’s d = 0.35), need to control thoughts (MD = 1.29, p > .001, Cohen’s d = 0.38) and cognitive self-consciousness (MD = 0.74, p > .001, Cohen’s d = 0.23).”

14. I think the section regression analysis is not easy to follow linguistically as links are not clear.

Attending to your recommendation, regression section has been modified as follows:

“Table 3 shows results of multiple regression analyses while Tables 4 and 5 show the model coefficients for each variable. The analyses revealed that sex and age were significantly related to cognitive reappraisal (explaining a 0.4% of the variance), expressive suppression (14.2%), assertiveness (0.7%), metacognitions (3.9%), positive beliefs about worry (5.2%), need to control thoughts (5.5%) and cognitive self-consciousness (6.6%), with p < .05 in all cases. Moreover, the inclusion of the rMEQ scores in the equation significantly increased the explained variance for cognitive reappraisal (0.5%), expressive suppression (0.3%), assertiveness (0.6%), metacognitions (0.8%), negative beliefs of uncontrollability and danger (0.9%), cognitive confidence (0.7%), need to control thoughts (0.3%) and cognitive self-consciousness (0.5%), with p < .05 in all cases.”

15. Also: was the increase from the first to the second model significant?

Significant increases from the first to the second model was observed for cognitive reappraisal, expressive suppression, assertiveness, metacognitions, negative beliefs of uncontrollability and danger, cognitive confidence, need to control thoughts and cognitive self-consciousness. Regarding your comment, the fact that the inclusion of the rMEQ scores in the equation resulted in a significantly increase of the explained variance was added to this section:

“Moreover, the inclusion of the rMEQ scores in the equation resulted in a significantly increase of the explained variance for cognitive reappraisal…”

16. Table 2 should be ordered the same way as the results section.

According to your recommendation, Table 2 has been ordered the same way as the results section

17. Table 4 und 5 are named alike. Aren’t the information in Table 5 subscales of metacognitive beliefs?

Attending to your comment, tables 4 and 5 have been renamed as follows: 

“Table 4. Multiple linear regression model coefficients for emotion regulation strategies, assertiveness and MCQ-30 total score.”

“Table 5. Multiple linear regression model coefficients for the different metacognitions assessed through MCQ-30.”

18. Please provide english explanations for the variables in the .sav-file on OSF.

The file with English explanations for the variables can be found at https://osf.io/nh5c4

---

## [Decision Letter · Decision Letter 2]

25 Feb 2020

Circadian typology is related to emotion regulation, metacognitive beliefs and assertiveness in healthy adults

PONE-D-19-23969R2

Dear Dr. Antúnez,

We are pleased to inform you that your manuscript has been judged scientifically suitable for publication and will be formally accepted for publication once it complies with all outstanding technical requirements.

With kind regards,

Angel Blanch, Ph.D.

Academic Editor

PLOS ONE

Additional Editor Comments (optional):

Reviewers' comments:

Reviewer's Responses to Questions

**Comments to the Author**

1. If the authors have adequately addressed your comments raised in a previous round of review and you feel that this manuscript is now acceptable for publication, you may indicate that here to bypass the “Comments to the Author” section, enter your conflict of interest statement in the “Confidential to Editor” section, and submit your "Accept" recommendation.

Reviewer #1: All comments have been addressed

2. Is the manuscript technically sound, and do the data support the conclusions?

Reviewer #1: Yes

3. Has the statistical analysis been performed appropriately and rigorously? 

Reviewer #1: Yes

4. Have the authors made all data underlying the findings in their manuscript fully available?

Reviewer #1: Yes

5. Is the manuscript presented in an intelligible fashion and written in standard English?

Reviewer #1: Yes

6. Review Comments to the Author

Reviewer #1: (No Response)

7. PLOS authors have the option to publish the peer review history of their article (what does this mean?). If published, this will include your full peer review and any attached files.

Reviewer #1: No

---

## [Editor Report · Acceptance letter]

2 Mar 2020

PONE-D-19-23969R2 

Circadian typology is related to emotion regulation, metacognitive beliefs and assertiveness in healthy adults 

Dear Dr. Antúnez:

I am pleased to inform you that your manuscript has been deemed suitable for publication in PLOS ONE. Congratulations! Your manuscript is now with our production department. 

With kind regards,

on behalf of

Dr. Angel Blanch 

Academic Editor

PLOS ONE